# Immunomodulatory Responses of Subcapsular Sinus Floor Lymphatic Endothelial Cells in Tumor-Draining Lymph Nodes

**DOI:** 10.3390/cancers14153602

**Published:** 2022-07-24

**Authors:** Eliane Sibler, Yuliang He, Luca Ducoli, Viviane Rihs, Patrick Sidler, Claudia Puig-Moreno, Jasmin Frey, Noriki Fujimoto, Michael Detmar, Lothar C. Dieterich

**Affiliations:** 1Institute of Pharmaceutical Sciences, Swiss Federal Institute of Technology (ETH) Zürich, 8093 Zürich, Switzerland; eliane.sibler@pharma.ethz.ch (E.S.); yuliang.he@pharma.ethz.ch (Y.H.); lducoli@stanford.edu (L.D.); virihs@student.ethz.ch (V.R.); patricksidler@hotmail.com (P.S.); cpuigmoreno@student.ethz.ch (C.P.-M.); jasmin.frey@pharma.ethz.ch (J.F.); 2Department of Dermatology, Shiga University of Medical Science, Otsu 520-2192, Japan; noriki@belle.shiga-med.ac.jp

**Keywords:** cancer, melanoma, scRNA-seq, lymph node, lymphatic endothelial cells, macrophage, subcapsular sinus

## Abstract

**Simple Summary:**

Lymph nodes (LNs) are essential for the activation of immune responses against tumors. LNs consist of immune cells and stromal cells, including lymphatic endothelial cells (LECs), that closely interact with each other. We performed single-cell RNA sequencing of LECs residing in tumor-draining LNs compared to normal LNs, to investigate their responses to tumor-derived signals. We found that a specific subset of LECs lining the floor of the subcapsular sinus, where the afferent lymph enters the LN, showed the most drastic changes in gene expression. Many of the upregulated genes were associated with inflammation and vessel growth. Furthermore, we found that several upregulated genes, including podoplanin, mediate adhesion of macrophages to LN LECs. Consequently, deletion of podoplanin on LECs reduced the number of LN macrophages in vivo. Our study shows that tumor-derived signals induce changes in LN LECs that may influence the tumor immune response.

**Abstract:**

Tumor-draining lymph nodes (LNs), composed of lymphocytes, antigen-presenting cells, and stromal cells, are highly relevant for tumor immunity and the efficacy of immunotherapies. Lymphatic endothelial cells (LECs) represent an important stromal cell type within LNs, and several distinct subsets of LECs that interact with various immune cells and regulate immune responses have been identified. In this study, we used single-cell RNA sequencing (scRNA-seq) to characterize LECs from LNs draining B16F10 melanomas compared to non-tumor-draining LNs. Several upregulated genes with immune-regulatory potential, especially in LECs lining the subcapsular sinus floor (fLECs), were identified and validated. Interestingly, some of these genes, namely, podoplanin, CD200, and BST2, affected the adhesion of macrophages to LN LECs in vitro. Congruently, lymphatic-specific podoplanin deletion led to a decrease in medullary sinus macrophages in tumor-draining LNs in vivo. In summary, our data show that tumor-derived factors induce transcriptional changes in LECs of the draining LNs, especially the fLECs, and that these changes may affect tumor immunity. We also identified a new function of podoplanin, which is expressed on all LECs, in mediating macrophage adhesion to LECs and their correct localization in LN sinuses.

## 1. Introduction

Lymph nodes (LNs) are essential sites for the activation of adaptive immune responses, as they orchestrate encounters and interactions between antigens, antigen-presenting cells, and lymphocytes. LN stromal cells, including lymphatic endothelial cells (LECs), are indispensable to the highly defined spatial organization of LNs and the regulation of leukocyte activities for optimal immune activation [1,2]. For instance, LN-resident LECs have been reported to maintain peripheral immune tolerance at steady-state [3,4], to regulate lymphocyte migration and survival via sphingosine-1-phosphate (S1P) [5,6], and to provide survival and differentiation signals for sinusoidal macrophages [7,8].

Tumor-draining LNs undergo massive enlargement and remodeling, which correlates with subsequent LN metastasis and poor patient outcome in various cancer types [9,10,11]. Of note, these changes may occur before metastatic colonization of the LNs, indicating the involvement of tumor-derived factors, such as extracellular vesicles (EVs) and soluble growth factors carried by afferent lymph [12,13,14,15]. Melanoma-derived EVs have been shown to interact directly with LN-resident LECs [16,17,18], inducing lymphangiogenesis and subsequent metastasis [17,18]. LN LECs have also been implicated in tumor immune evasion and immunotherapy resistance by cross-presenting tumor-derived antigens [19] and by expressing the T cell-inhibitory molecule PD-L1 [20]. Yet, our overall knowledge of the phenotypic changes of LECs in response to tumor-derived signals and the consequences for LN microenvironment and tumor immunity are still incomplete.

LECs lining distinct LN sinuses differ in their gene expression and functions [21]. Recent single-cell RNA sequencing (scRNA-seq) studies addressed LEC heterogeneity in steady state and suggest three major LEC subsets that are conserved between mice and human, namely, ceiling LECs (cLECs) and floor LECs (fLECs) lining the subcapsular sinus and medullary LECs (mLECs) [22,23,24]. The fLECs represent an important interface between the lymph and the underlying cortex, and they regulate access of antigens and antigen-presenting cells to the conduit system and the B cell follicles. Furthermore, fLECs closely interact with sinusoidal macrophages and LN-resident dendritic cells (DCs) that scavenge antigen from the lymph. Comparatively, mLECs have been reported to be the major regulators of peripheral tolerance owing to their highest expression of peripheral self-antigen tyrosinase among all LN LECs [25].

By using microarray or bulk RNA sequencing following flow sorting, we and others have collectively mapped transcriptional responses of LN LECs to inflammatory signals, virus infection, and tumor growth [26,27,28]. However, LN LEC subtype-specific responses were not differentiated in these studies. More recently, we and others investigated the transcriptional changes in LN LECs upon dermatitis at single-cell resolution. fLECs responded in the most dynamic manner to pathologic stimuli derived from upstream tissues, which may alter the access of lymph-borne antigen-presenting cells, antigens, and other soluble factors to the LN parenchyma [24,29]. Here, we transcriptionally characterized LN LECs in tumor-draining LNs at the single-cell level and again observed dynamic responses, particularly in fLECs. Furthermore, by functional studies in vitro and in vivo, we identified several tumor-induced molecules that mediate interactions between LN LECs and sinusoidal macrophages.

## 2. Materials and Methods

### 2.1. Animals

Mice carrying loxP sites flanking exon 2 of the *Pdpn* gene [30] were crossed with Prox1-CreER^T2^ mice [31] to generate Cre+ and Cre− littermates. For lymphatic *Pdpn* deletion, mice were i.p. injected 5 times with tamoxifen (Sigma-Aldrich, Burlington, MA, USA, 50 mg/kg) in sunflower oil. Ackr4-GFP reporter mice [32] (kindly provided by Dr. Cornelia Halin, ETH Zürich) and C57Bl/6N mice were bred in-house in an SPF facility, and the experimental animal procedures were approved by the responsible ethics committee (Kantonales Veterinäramt Zürich, licenses 5/18 and 101/21).

### 2.2. Tumor Studies

A total of 200,000 B16F10-luc2 cells (Caliper, Hopkinton, MA, USA) in 20 µL PBS were implanted intradermally on both sides of the shaved back skin of age- and sex-matched mice (8–12 weeks old), and tumor growth was monitored by caliper measurements. On day 14 after tumor implantation, mice were sacrificed and the tumor-draining axillary and inguinal LNs were collected, embedded, and frozen in O.C.T. compound or processed for flow cytometry.

### 2.3. scRNA-seq of LN LECs

Tumor-draining inguinal LNs from mice bearing B16F10-luc2 melanomas (*N* = 3) were harvested on day 14 for sorting of LECs by FACS and processed in parallel with our previous study for scRNA-seq [18]. ScRNA-seq data of the naive control is accessible at ArrayExpress under the accession number E-MTAB-10434. scRNA-seq analyses were performed as previously described [18]. Briefly, quality filtering was performed with the scran package v1.4.5 [33]; cells with library size 3 median absolute deviations (MADs) away from the median or with feature size and mitochondrial contents 3 MADs above the median were dropped as outliers. Genes expressed in at least 15% of the cells were grouped in accordance with their count–depth relationship using SCnorm v0.99.7 [34], which applied a quantile regression within each group to estimate scaling factors and normalize for sequencing depth. Cells with detected *Ptprc* (CD45) expression were removed prior to downstream analyses. The top 2000 variable features were identified in the naive and tumor-draining LN LEC datasets and were subsequently integrated using the ‘FindIntegrationAnchors’ and ‘IntegrateData’ functions in the Seurat package v3.1.2 [35]. Unsupervised clustering was performed on the integrated dataset and visualized with Uniform Manifold Approximation and Projection (UMAP) [36]. Differentially expressed (DE) genes in each respective LN LEC subtype between naive and tumor conditions were identified by the ‘FindMarkers’ function (min.pct = 0.20, logfc.threshold = 0) using the MAST test [37], and filtered for logfc > 0.25, p_val_adj < 0.05. Expression patterns of selected markers were plotted by the ‘VlnPlot’ and ‘DotPlot’ functions.

### 2.4. Gene Ontology (GO), Transcription Factor Motif Analysis, and Comparison to Previously Published LN LEC Gene Expression Datasets

GO analysis was performed with DAVID bioinformatics resource (2021 update) [38,39]. Enrichment of GO terms for biological processes was analyzed with the functional annotation tool using all genes expressed in the respective subset of LECs that were not DE between the two conditions as a custom background. Transcription factor motif analysis for genes upregulated in fLECs from tumor-draining LNs was performed with ‘findMotifs.pl’ (-start -2000 -end 1000) from HOMER (v4.11.1) [40].

To compare the genes upregulated in tumor-draining LN LECs with previously published gene expression datasets of LN LECs exposed to various types of inflammation, we reanalyzed a dataset of LN LECs from day 6 after Herpes simplex virus (HSV) infection [26] and during inflammation induced by ovalbumin injection into mice after adoptive transfer of OT-1 T cells [27] using Geo2R [41], from which DE genes were defined by log2FC > 1 and adjusted *p* value < 0.05. In addition, DE gene lists (logfc > 0.25, adjusted *p* value < 0.05) from a scRNA-seq study of LN LECs in imiquimod-induced psoriasiform skin inflammation were retrieved from the Appendix A [29].

### 2.5. RNA Velocity Analysis

RNA velocity analysis was performed with velocyto.py as previously described [42]. Briefly, spliced and unspliced counts were calculated with the ‘run-smartseq2‘ function. RNA velocity of protein-coding genes was then estimated with the function ‘gene.relative.velocity.estimates‘ from the velocyto.R package using default settings except for parameters kCells = 5 and fit.quantile = 0.5 [42]. We restricted our analysis to protein-coding genes to avoid nuclear or cytoplasmic RNA retention biases.

### 2.6. Flow Cytometry Analysis of Tumor-Draining LNs

Tumor-draining LNs were pooled and digested either sequentially with 1 mg/mL and 3.5 mg/mL collagenase type IV (Thermo Fisher Scientific, Waltham, MA, USA) to enrich for LN stromal cells, as previously described [20], or directly with 3.5 mg/mL collagenase type IV in DMEM (Thermo Fisher Scientific) supplemented with 2% FBS (Thermo Fisher Scientific) and 1.2 mM CaCl_2_. AccuCheck counting beads (Thermo Fisher Scientific) were added for absolute cell number determination, and anti-CD16/CD32 (101302, BioLegend, San Diego, CA, USA, 1:100) was used to block Fc receptors.

To analyze podoplanin and CD200 in tumor-draining vs. control LNs in Ackr4-GFP mice, cells were stained with CD45-PacificBlue, CD31-PerCp/Cy5.5, Podoplanin-Pe/Cy7, Mrc1-Pe, CD44-BV650, and CD200-Pe/Dazzle594. To analyze LN stromal cells in Pdpn^fl/fl^ x Prox1-CreER^T2^ mice and control littermates, we used CD45-Apc/Cy7, CD31-PerCp/Cy5.5, Podoplanin-Pe/Cy7, CD41-Pe, Ter119-Fitc, and ESAM, followed by donkey anti-goat-Alexa647. To analyze LN macrophages, we used CD11b-BV605, CD169-Pe/Cy7, F4/80-Alexa647, MHCII-PerCp, CD80-Fitc, and CD86-Pe. Details about all antibodies are listed in Appendix A. Zombie-Aqua (BioLegend, 1:500) was used for live/dead staining. All data were recorded on a Fortessa instrument (BD Biosciences, Franklin Lakes, NJ, USA).

### 2.7. Immunofluorescence Staining of LN Sections and Image Analysis

LNs were embedded in O.C.T. compound, snap frozen in liquid nitrogen, and stored at −80 °C until preparation of 7 µm-thick cryosections. For staining, sections were first fixed with ice-cold acetone and methanol, dried and rehydrated in PBS, and subsequently blocked with blocking solution (PBS with 5% donkey serum, 0.3% Triton-X100, 0.2% BSA, and 0.05% NaN_3_) before primary antibodies against LYVE-1, BST2, tenascin, ANXA2, CD44, MRC-1, CCL21, CD9, VCAM-1 and CD169 diluted in blocking solution were applied. After extensive washing, slides were incubated with corresponding secondary antibodies (donkey anti-rat-Alexa488, anti-rabbit-Alexa488, anti-rabbit-Alexa594, anti-goat-Alexa488 and anti-goat-Alexa594) together with Hoechst33342 for nuclear counterstaining, mounted, and imaged using an AxioScope 2 mot plus microscope (Carl Zeiss, Oberkochen, Germany). Details about all antibodies are listed in Appendix A. We took 2–3 images per LN covering subcapsular and/or medullary sinuses. To confirm BST2 and tenascin upregulation in fLECs, the LYVE-1+ area was selected by manual threshold and the average intensity of BST2 and Tenascin within this area was measured using Fiji v2.3.0 [43]. The macrophage coverage of lymphatic sinuses was determined by selecting both the CD169+ and LYVE-1+ areas using manual thresholds and measuring the overlap between them.

### 2.8. Isolation and Culture of Primary LN LECs

Primary mouse LN LECs were isolated as previously described [28]. Briefly, popliteal, inguinal, axillary, brachial, and auricular LNs were isolated from 8–12-week-old animals; torn; and digested in RPMI medium (Thermo Fisher Scientific) containing 0.25 mg/mL Liberase TL (Roche, Basel, Switzerland) and 1 mg/mL DNase I (Roche) at 37 °C for 1 h. After 15 min, the LNs were cut with scissors, and thereafter further disrupted by pipetting every 10–15 min. The cell suspension was passed through a 70-µm cell strainer, centrifuged, and resuspended in full LN LEC medium (αMEM (no nucleosides, Thermo Fisher Scientific), 10% FBS, 1% penicillin-streptomycin (Thermo Fisher Scientific), and 1% L-glutamine (Thermo Fisher Scientific)). Cells were cultured at 37 °C and 5% CO_2_ in dishes precoated with 10 µg/mL collagen (PureCol, Advanced BioMatrix, Carlsbad, CA, USA) and 10 µg/mL human fibronectin (Merck, Darmstadt, Germany) in PBS. Nonadherent cells were washed away with warm PBS on day 1 and day 3. Once the cells reached about 80% confluency, they were detached with accutase (Sigma-Aldrich) and purified with CD31 magnetic beads (Miltenyi Biotec, Bergisch Gladbach, Germany) according to the manufacturer’s instructions. Purified LECs were cultured in full medium on coated plates and used for experiments in passages 3–5.

### 2.9. Induction of Target Genes by Cytokines

LN LECs were seeded in full medium and treated with 10 ng/mL TNFα (R&D Systems, Minneapolis, MN, USA), 100 ng/mL IFNγ (PeproTech, London, UK), 10 ng/mL TNFα + 100 ng/mL IFNγ, or 10 ng/mL TGFβ (BioLegend) for 4 h, 18 h, and 48 h. For qPCR, the cells were lysed and RNA was isolated with the NucleoSpin RNA kit (NucleoSpin RNA, Macherey-Nagel, Düren, Germany), according to the manufacturer’s instructions. Reverse transcription was performed with the High Capacity cDNA Reverse Transcription Kit (Thermo Fisher Scientific) and qPCR with a 7900HT Fast Real-Time PCR System (Thermo Fisher Scientific) using PowerUp SYBR Green Master Mix (Thermo Fisher Scientific). Primers were designed for *Pdpn*, *Cd200*, *Bst2*, *Tnc*, and *Rplp0* (Appendix A).

For flow cytometry, the cells were enzymatically detached and stained with CD31-APC, CD200-Pe/Dazzle594, podoplanin-Pe/Cy7, and BST2. Staining with Zombie NIR (423106, BioLegend, 1:500) and secondary donkey anti-rat IgG-Alexa488 antibody where necessary were conducted in a second step. Details about all antibodies are listed in Appendix A. Cells were fixed with Cytofix/Cytoperm solution (BD Biosciences) before analysis. Data were recorded on a Cytoflex 2L (Beckman Coulter, Brea, CA, USA) and analyzed with FlowJo (BD Biosciences, version 10).

### 2.10. Viability Assay

LN LECs were seeded sparsely in black, clear-bottom plates (Corning Incorporated, Corning, NY, USA) and left to attach for 4 h. Full medium was replaced with starvation medium (0.5% FBS, 1% penicillin-streptomycin, 1% L-glutamine in αMEM) and the cells were incubated at 37 °C overnight. Then, the medium was replaced by either fresh starvation or full medium and LECs were treated with 10 µg/mL blocking antibody against CD200, tenascin or BST2 or with corresponding isotype controls (Appendix A) for 48 h. Then, the medium was removed, 0.1 mg/mL methylumbelliferone (Sigma-Aldrich) was added to the cells for 1 h at 37 °C, and the fluorescence was measured using a plate reader (SpectraMAX GEMINI EM, Molecular Devices, San Jose, CA, USA) with excitation 355 nm and emission 469 nm.

### 2.11. Scratch Assay 

LN LECs were seeded on 96-well plates and incubated until a confluent layer was reached, followed by mitomycin-C treatment (2 µg/mL, Sigma-Aldrich) for 2 h. Thereafter, a scratch was inflicted in each well using a scratch brush (V&P Scientific, San Diego, CA, USA). The cells were washed and 10 µg/mL blocking antibodies against CD200, tenascin, or BST2, or the corresponding isotype controls in either starvation or full LN LEC medium were added. Images were taken with an Axiovert 200M microscope and AxioCam MRm (Carl Zeiss) with 10× magnification at 0 h and 16 h. Scratch closure was quantified using TScratch [44].

### 2.12. Macrophage Adhesion Assay 

Confluent LN LEC monolayers were treated with 10 ng/mL TNFα or medium overnight. Medium was then replaced with macrophage medium (RPMI1640 + GlutaMax (Thermo Fisher Scientific), 10% FCS, 1% penicillin-streptomycin, and 0.05 mM 2-mercaptoethanol) containing 10 µg/mL blocking antibodies against CD200, tenascin, or BST2, or corresponding controls (Appendix A). LN LECs were incubated with antibodies at 37 °C for 30 min. Macrophages were differentiated from bone marrow cells harvested from the tibias and femurs of C57Bl/6N mice. Bone marrow cells were cultured in macrophage medium with 50 ng/mL murine M-CSF (PeproTech) for 7 days. A total of 50,000 macrophages were then added to the LECs and incubated for 30 min in presence of the blocking antibodies. Then, nonadherent cells were removed and wells were washed three times with PBS. Adherent cells were detached with trypsin, collected, and stained with rat anti-mouse CD11b-FITC antibody (Appendix A). Data were recorded on a Cytoflex S (Beckman Coulter) instrument. 

### 2.13. Statistical Analysis

Prism (GraphPad Software, San Diego, CA, USA, Version 9) was used for statistical analyses and to plot graphs. Test details are indicated in the corresponding figure legends. All bioinformatic analyses were performed with R v3.6.3.

## 3. Results

### 3.1. scRNA-seqReveals Persistence of LN LEC Subsets in Tumor Conditions

In order to characterize the transcriptional landscape of LECs in tumor-draining LNs, we isolated inguinal LNs from control and B16F10 tumor-bearing mice. These LNs were digested, and LECs were sorted by FACS and subjected to scRNA-seq with the Smart-seq2 approach (Figure 1a) [45]. A total of 225 high-quality cells from control and 356 from tumor-draining LNs were included in the downstream analysis and their LEC identity could be confirmed by the expressions of endothelial cell markers Cd31 (*Pecam1*) and VE-cadherin (*Cdh5*) as well as lymphatic markers *Prox1* and Vegfr3 (*Flt4*) (Figure 1b). These cells clustered into three distinct subsets (Figure 1c), identified as fLECs, cLECs, and mLECs (Figure 1d) by their expression of previously described markers such as *Madcam1* (mucosal vascular addressin cell adhesion molecule 1), *Ackr4* (atypical chemokine receptor 4), and *Mrc1* (mannose receptor, C type 1), respectively (Figure 1b) [22,46]. Notably, LECs from tumor-draining LNs maintained their expression of these and other known subset markers (Figure 1b). Furthermore, cells from control and tumor-draining LNs were well-integrated and the relative abundance of the three subsets was comparable between the two conditions (Appendix A). Together, these data indicate that the general subset identities of LECs are not altered in tumor-draining LNs.

### 3.2. Podoplanin, BST2, CD200, and Tenascin Are Upregulated in fLECs in Tumor-Draining LNs

To investigate the effect of tumor-associated signals on the draining LNs, we performed DE analysis within each LEC subset (Appendix A). While the numbers of up- compared with downregulated genes were similar within each subset, by far the most dramatic transcriptional change was observed in fLECs (Figure 2a and Appendix A). GO analysis highlighted terms related to angiogenesis, cell migration, cell proliferation, and inflammation (Appendix A), while transcription factor motif enrichment analysis additionally suggested an activation of interferon (IRF1) and TGF (SMAD2) signaling in tumor-draining LN fLECs (Appendix A). Furthermore, RNA velocity analysis, a computational method to predict future mRNA dynamics [42], suggested a significant difference between the control and tumor-draining LECs in the fLEC (and mLEC) compartment (Appendix A), suggesting that these cells actively adapt their transcriptome in response to primary tumor growth, whereas cLECs appeared comparably inert.

Next, we selected four genes upregulated in fLECs in tumor-draining LNs, namely, *Pdpn* (podoplanin), *Cd200*, *Bst2* (tetherin, CD317), and *Tnc* (tenascin), for confirmation on the protein levels. These genes, which have previously been reported to function in (lymph-) angiogenesis and immunity [47,48,49,50,51,52,53,54,55], were induced in LN LECs in the context of B16F10 melanoma growth or B16F10-derived EVs [18,28], and/or were associated with several of the top-deregulated GO terms (Appendix A). The mucin-like type I transmembrane protein podoplanin is a classic LEC marker gene highly expressed on LECs, including all subsets of LN LECs. Nonetheless, podoplanin was significantly upregulated only in fLECs in B16F10-draining LNs, both on the transcript and the protein levels, as determined by flow cytometry (Figure 2b,c and Appendix A). In contrast, the immune-regulatory molecule CD200 [49,50,51,52] was expressed mainly by fLECs at steady-state, and further induced in tumor-draining LNs (Figure 2d,e). The antiviral protein BST2 [53] and the extracellular matrix protein tenascin [52] were again broadly expressed by all three LEC subsets, but nonetheless showed the strongest elevation in fLECs in tumor-draining LNs (Figure 2f–i). Interestingly, BST2 upregulation in fLECs was already detectable by immunofluorescence staining at day 6 after tumor implantation, whereas tenascin was only induced at late stages, suggesting varying kinetics of tumor-induced responses by fLECs (Appendix A).

### 3.3. Podoplanin, BST2, and CD200 Are Induced by Proinflammatory Cytokines in Primary LN LECs In Vitro

To identify potential mechanisms or signaling pathways regulating the expression of these genes, we isolated primary mouse LN LECs, stimulated them with cytokines involved in inflammatory and TGF-associated signaling pathways that might be active in tumor-draining LNs, and assessed their expression by qPCR and flow cytometry. Compared with untreated cells, stimulation with a combination of IFNγ and TNFα resulted in podoplanin upregulation on the mRNA level as early as 4 h after stimulation, and, with some delay, on the protein level (Figure 3a,b). In contrast, the expression of CD200 mRNA was rapidly but transiently upregulated by this combination, while BST2 was most strongly induced by IFNγ alone, especially at the 24 h timepoint (Figure 3a,b). In both cases, upregulation on the protein levels was sustained up to 48 h after stimulation. *Tnc*, however, could not be induced by any of the cytokines we used (Figure 3a). These results indicate that the upregulation of podoplanin, BST2, and CD200 in LN LECs might be mediated by inflammatory cytokines in tumor-draining LNs.

### 3.4. Podoplanin, BST2, CD200, and Tenascin Do Not Affect Lymphangiogenic Responses In Vitro

Next, we sought to investigate these genes functionally. We first hypothesized that podoplanin could be functionally involved in lymphangiogenesis or lymphatic expansion. Therefore, we generated a conditional, lymphatic *Pdpn*-knockout mouse model by crossing Pdpn^fl/fl^ with Prox1-CreER^T2^ mice. Indeed, after tamoxifen treatment of Pdpn^fl/fl^ x Prox1-CreER^T2^ mice and isolation of primary LN LECs, these cells had lost expression of podoplanin to a large extent compared with cells derived from Cre− littermate controls (Appendix A). However, genetic deletion of *Pdpn* had no major effect on the viability and migratory capacity of LN LECs in vitro (Appendix A). Similarly, antibody-mediated blockade of BST2, CD200, and tenascin in cultured LN LECs derived from wildtype mice had no measurable effect on cell viability and migration (Appendix A).

### 3.5. Podoplanin Promotes Adhesion of Macrophages to LN LECs In Vitro, While BST2 and CD200 Reduce It

CD169+ sinusoidal macrophages tightly interact with LN LECs, attaching to the luminal LEC surface and extending processes across the LEC layer [56]. However, the molecular interactions mediating adhesion between sinusoidal macrophages and LN LECs are not fully understood. Thus, we next aimed to investigate whether podoplanin, BST2, CD200, or TNC might be involved in this process. Interestingly, adhesion of bone-marrow-derived macrophages to cultured LN LECs derived from Pdpn^fl/fl^ x Prox1-CreER^T2^ mice was reduced compared with control in basal conditions (Figure 4a). In line with our previous data, TNFα treatment of cultured LECs further induced podoplanin expression in Cre− control cells (Appendix A). However, macrophage adhesion was not affected by podoplanin deletion in this case, most likely due to upregulation of other adhesion molecules by acute TNFα stimulation that compensate for the lack of podoplanin, such as ICAM-1, VCAM-1, or CD31. On the other hand, pretreatment of wildtype LN LECs with antibodies blocking BST2 or CD200 resulted in a significantly increased number of adhered macrophages compared with the isotype control (Figure 4b,c). This enhancing effect was observed both under basal conditions and following stimulation of LN LECs with TNFα. Antibody blockade of tenascin had no impact on macrophage adhesion compared with the isotype control (Figure 4d). In summary, these data show that podoplanin promotes macrophage adhesion to LECs in vitro, while CD200 and BST2 inhibit it.

### 3.6. Podoplanin Is Required for Appropriate Macrophage Localization in Medullary Sinuses In Vivo

As podoplanin expression was required for optimal macrophage adhesion to LN LECs in vitro, we sought to confirm this hitherto unknown function of podoplanin in tumor-draining LNs in vivo. To this end, Pdpn^fl/fl^ x Prox1-CreER^T2^ mice and Cre− littermate controls were treated with tamoxifen and subsequently challenged by orthotopic implantation of B16F10 melanoma cells. Tumor growth was followed over a course of 14 days before analysis of draining LNs (Figure 5a). Lymphatic podoplanin deletion did not affect primary tumor growth, LN weight, or content of red blood cells in LNs (Appendix A), in contrast to previous findings where lymphatic *Pdpn* deletion right after birth increased LN weight and red blood cells in LNs of adult mice [57]. CD41 (Itga2b) expressed in fLECs and mLECs [22,58], and ESAM expressed in blood endothelial cells (BECs) [59] were used to differentiate LECs in flow cytometry (Figure 5b) and to confirm strongly reduced podoplanin expression in LN LECs of Pdpn^fl/fl^ x Prox1-CreER^T2^ mice compared with control littermates, whereas podoplanin expression in fibroblastic reticular cells (FRCs) and BECs was not affected (Figure 5c,d). Additionally, we found no differences in the numbers of BECs, LECs, and FRCs, in line with our observation that podoplanin deletion did not affect lymphangiogenesis in vitro (Figure 5e). Additionally, the identity of the major LN LEC subsets as well as their expression of known podoplanin-interacting molecules such as CD44, CCL21, and CD9 [60] or the adhesion-molecule VCAM-1 seemed unaffected (Appendix A).

Next, we turned our attention to LN macrophages. Staining of tumor-draining LN sections for the fLEC/mLEC marker LYVE-1 and the sinusoidal macrophage marker CD169 revealed that the density of CD169+ macrophages in medullary sinuses was reduced in Pdpn^fl/fl^ x Prox1-CreER^T2^ mice compared with Cre− control littermates, whereas the coverage of the subcapsular sinus by CD169+ macrophages was comparable in both groups (Figure 5f,g). In line with the immunofluorescence data, the absolute number and the relative frequency of medullary sinus macrophages (MSMs) tended to be reduced in tumor-draining LNs of Pdpn^fl/fl^ × Prox1-CreER^T2^ mice (Figure 5h–j), whereas the expression of costimulatory molecules by these macrophages was not altered (Appendix A). In conclusion, our data suggest that lymphatic podoplanin acts as a macrophage adhesion receptor to maintain or expand sinusoidal macrophages in tumor-draining LNs.

## 4. Discussion

Tumor-draining LNs are crucial for tumor immunity and for immunotherapy responses [61,62,63]. LN LECs are an important component of the LN stroma that can impair T cell responses via expression of PD-L1 and presentation of tumor antigens [18,19,20]. However, the low abundance of LECs within the total cell population in LNs [64,65] complicates in-depth investigations of LN LEC phenotypes. Here, we isolated CD31+ podoplanin+ LN LECs and devised the transcriptional responses of individual cells to primary tumor growth. Intriguingly, in line with a previous study on the remodeling of LN LECs in imiquimod-induced psoriasiform skin inflammation, we observed the most drastic changes in the fLECs from tumor-draining LNs compared with the other compartments [29]. Probably, this is because fLECs are directly exposed to molecular signals present in the afferent lymph, such as cytokines, growth factors, or EVs, and dynamically adapt to such signals to regulate immune cell and antigen access to the LN cortex. However, we cannot exclude that the identification of DE genes in the cLEC and mLEC subsets in our study might have been restricted by the relatively low frequency of these two subpopulations.

Previously, we showed that there is massive lymphatic expansion and proliferation of LECs in tumor-draining LNs in the B16F10 melanoma model [28]. Congruently, the genes upregulated in tumor-draining LN fLECs, including several transcription factors that regulate lymphatic differentiation and lymphangiogenesis (*Prox1*, *Klf4*, *Sox7*, and *Yap1*) [66], were significantly enriched for proliferation-, migration-, and angiogenesis-related GO terms. We also noted indications of an inflammatory response and TGF signaling in tumor-draining LN fLECs, including an upregulation of *Nfkb1* expression (Appendix A) and an enrichment of IRF1- and SMAD2-binding motifs (Appendix A). Yet, the phenotype of tumor-draining LN fLECs only marginally overlapped with previously published datasets of LN LECs isolated from acute inflammatory conditions [26,27,29], suggesting that tumor-associated inflammation in draining LNs is very distinct compared to those conditions (Appendix A).

As a downstream target of Prox1, podoplanin is required for lymphatic specification and separation from the blood vasculature during embryonic development [67,68]. Postnatal lymphatic-specific deletion of podoplanin, without affecting postnatal lymphatic development and patterning or LN organization, can result in retrograde blood filling of the lymphatic system and increased red blood cell numbers in peripheral LNs [56]. However, this was not observed in the present study, likely due to a shorter time span following podoplanin deletion in the lymphatic endothelium and/or reduced susceptibility of adult mice. Furthermore, although podoplanin has been implicated in pathological lymphangiogenesis via interaction with galectin-8 [47,48], we found no obvious effects on tumor-induced LN lymphangiogenesis in mice lacking lymphatic podoplanin expression. Besides mediating LEC-DC interactions via C-type lectin-like receptor 2 (CLEC-2) [69], we herein found that podoplanin also regulates the adhesion of macrophages as lymphatic podoplanin deletion impaired macrophage adhesion to LEC monolayers in vitro and reduced the density of macrophages in medullary sinuses of tumor-draining LNs in vivo. Recently, we transcriptionally characterized LN-resident macrophages and revealed that CLEC-2 is largely absent from MSMs [70], suggesting the participation of alternative podoplanin receptors in mediating MSM interactions, such as CD44 or the recently identified CD177 [71] that are more highly expressed in MSMs at steady-state [70]. In addition, podoplanin might mediate sinusoidal macrophage adhesion to LECs through attached sialoglycans binding to CD169 [72].

CD200 is broadly expressed on endothelial, hematopoietic, and tumor cells [73] and can act as an immune-inhibitory molecule to impair macrophage and DC activation [49,50]. Conversely, other studies have found that CD200 expression on tumor cells can reduce tumor growth and metastasis and increase the efficacy of T cell therapy [50,51]. These conflicting results may be due to several isoforms of the receptor CD200R, some of which appear to have activating or inhibitory functions [74,75]. BST2, on the other hand, has mostly been associated with antiviral responses [51] but might also tether tumor-derived EVs to LN LECs [18,76]. In line with earlier studies [77,78,79], we found that CD200 and BST2 could be induced in cultured LN LECs by TNFα and IFNγ, respectively. Moreover, they can both inhibit the adhesion of macrophages to LEC layers in vitro. Interestingly, a previous study investigating macrophage adhesion to a mouse brain endothelial cell line suggested that CD200-CD200R signaling reduces the ability of macrophages to adhere to endothelia [80]. In contrast, BST2 on BECs promotes the adhesion of monocytes in vitro [79], suggesting that BST2 may have different effects on cell–cell interactions depending on the cell types involved.

LN macrophages have emerged as important regulators of tumor immunity, promoting both CD8+ cellular and humoral immune responses in preclinical tumor studies [70,81,82]. Congruently, the density of CD169+ LN macrophages correlates with a better outcome in melanoma patients [83]. In our study, lymphatic podoplanin deletion and consequential reduction in MSMs did not impact on primary B16F10 tumor growth, which is in contrast to the complete deletion of both MSMs and subcapsular sinus macrophages (SSMs) using CD169-DTR mice that exacerbated primary tumor growth [82]. Nonetheless, macrophage amplification in LNs—for instance, via an induction of podoplanin expression in LECs or via blockade of CD200 or BST2—could help to activate endogenous as well as vaccine-induced tumor immunity [84].

## 5. Conclusions and Limitations

Here, we present a transcriptional map of LECs in tumor-draining LNs at the single-cell level. Our data demonstrate that LECs lining the floor of the subcapsular sinus, which express an immunity-associated gene signature and directly interact with soluble factors, EVs, and cells present in the afferent lymph, respond very dynamically to primary tumor-derived signals by upregulating many genes associated with (lymph-) angiogenesis and the regulation of immune responses, including *Pdpn*, *Cd200*, *Bst2*, and *Tnc*. Notably, we found that podoplanin, CD200, and BST2 regulate macrophage adhesion to LN LECs in vitro and lymphatic podoplanin was required to maintain macrophages in medullary sinuses but not in the subcapsular sinus of tumor-draining LNs in vivo. Together, our findings suggest that LEC subsets in tumor-draining LNs differentially respond to tumor-derived signals, affecting interactions with LN-resident macrophages that may alter tumor immunity. However, our study also has several limitations. For instance, due to our choice of the plate-based Smart-seq2 approach (in contrast to, e.g., a droplet-based approach), the overall number of LN LECs included in our dataset is relatively low, whereas the sequencing depth per cell is comparably high. Furthermore, we have analyzed LN LECs from a single mouse melanoma model, B16F10, which is known to be poorly immunogenic. Therefore, a broader screen of LEC phenotypes in tumor-draining LNs may unravel additional immune-regulatory pathways that could be exploited therapeutically in the future.

## Figures and Tables

**Figure 1 cancers-14-03602-f001:**
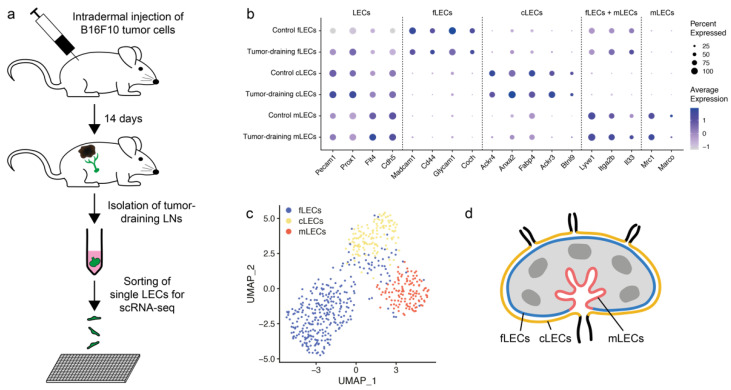
Single-cell RNA sequencing (scRNA-seq) shows persistence of lymphatic endothelial cell (LEC) subsets in lymph nodes (LNs) draining tumors. (**a**) Experimental workflow. B16F10 melanoma cells were injected intradermally into the flanks of C57Bl/6N mice. After 14 days, mice were sacrificed and tumor-draining inguinal LNs collected. LNs were digested enzymatically, the resulting single cell suspension stained for lymphatic markers, and individual LECs sorted by FACS into 384-well plates for scRNA-seq. (**b**) Dot plot showing expression of established LEC and subset marker genes in subcapsular sinus floor LECs (fLECs), subcapsular sinus ceiling LECs (cLECs) and medullary LECs (mLECs) from tumor-draining and control LNs. (**c**) UMAP visualization of all 581 LN LECs used for analysis, 225 from control and 356 from tumor-bearing mice. Cells are colored by cluster; fLECs (blue), cLECs (yellow), and mLECs (red). (**d**) Schematic illustration of LEC subtypes within the LN.

**Figure 2 cancers-14-03602-f002:**
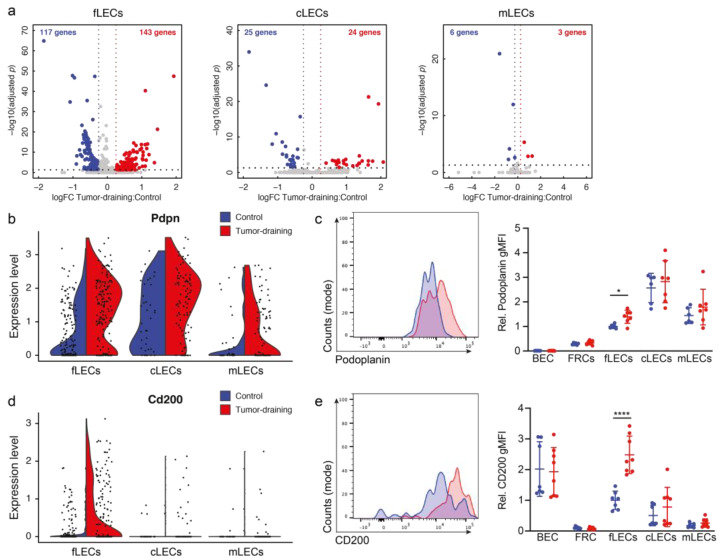
Transcriptional changes in LN LEC subsets in response to tumor growth. (**a**) Volcano plots showing differentially expressed (DE) genes for LECs of each subset. Significantly up- and downregulated genes and their number are shown in red and blue, respectively. (**b**) Violin plot of *Pdpn* expression in LECs from control (blue) and tumor-draining (red) LNs. (**c**) Representative histogram of podoplanin expression in fLECs (left) and quantification of podoplanin geometric mean fluorescence intensity (gMFI) in blood endothelial cells (BECs), fibroblastic reticular cells (FRCs), and LN LEC subsets in tumor-draining (red) compared with control LNs (blue). gMFI values were normalized to control fLECs in order to pool mice from two individual experiments (*N* = 6 control/7 tumor-draining). (**d**) Violin plot of *Cd200* expression in LECs from control (blue) and tumor-draining (red) LNs. (**e**) Representative histogram of CD200 expression in fLECs (left) and quantification of CD200 gMFI in BECs, FRCs, and LN LEC subsets in tumor-draining (red) compared with control LNs (blue). gMFI values were normalized to control fLECs in order to pool data from two individual experiments (*N* = 7 control/8 tumor-draining). (**f**) Violin plot of *Bst2* expression in LECs from control (blue) and tumor-draining (red) LNs. (**g**) Representative immunofluorescence staining of LYVE-1 and BST2 in control and tumor-draining LNs and quantification of relative BST2 fluorescence intensity within LYVE-1+ subcapsular sinus (*N* = 5/group). (**h**) Violin plot of *Tnc* expression in LECs from control (blue) and tumor-draining (red) LNs. (**i**) Representative immunofluorescence staining of LYVE-1 and tenascin in control and tumor-draining LNs and quantification of relative BST2 fluorescence intensity within LYVE-1+ subcapsular sinus (*N* = 5/group). White arrows indicate the floor of the subcapsular sinus. * *p* < 0.05, ** *p* < 0.01, **** *p* < 0.0001, unpaired *t*-test.

**Figure 3 cancers-14-03602-f003:**
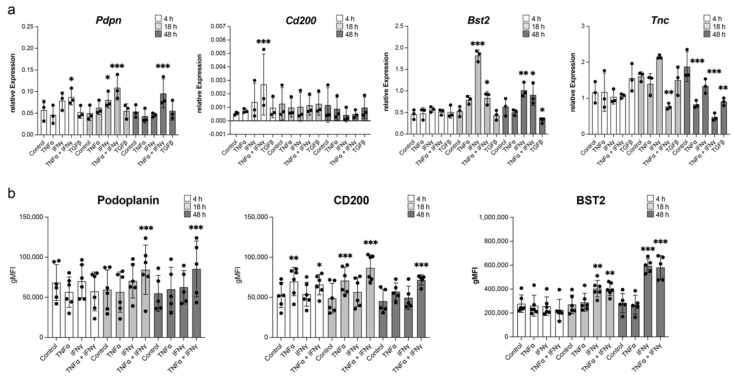
Induction of selected genes in primary LN LECs by cytokines. (**a**) Induction of *Pdpn*, *Cd200*, *Bst2*, and *Tnc* expression in LN LECs treated with cytokines for 4 h (white), 18 h (light-gray), or 48 h (dark-gray) measured by qPCR. Pooled data from three experiments, presented as relative expression normalized to *Rplp0* at each time point. Individual values, mean, and SD are shown. (**b**) Induction of podoplanin, CD200, and BST2 in LN LECs treated with cytokines for 4 h (white), 18 h (light-gray), or 48 h (dark-gray) measured by flow cytometry. Pooled data from six experiments, gMFI of each sample, mean, and SD are shown. * *p* < 0.05, ** *p* < 0.01, *** *p* < 0.001, RM one-way ANOVA with Geisser–Greenhouse correction and Dunnett’s multiple comparisons test comparing the means of each time-point to the respective control.

**Figure 4 cancers-14-03602-f004:**
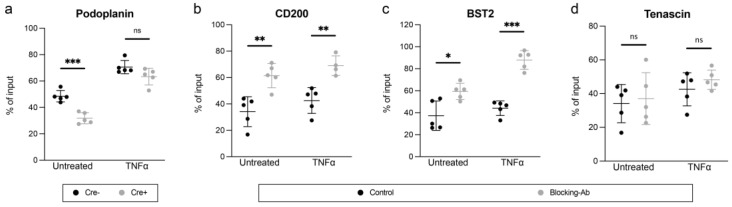
Podoplanin, CD200, and BST2 impact macrophage adhesion to primary LN LECs. (**a**–**d**) In vitro adhesion of macrophages to a LN LEC monolayer after podoplanin deletion (**a**) or antibody-mediated blockade of CD200 (**b**), BST2 (**c**), or tenascin (**d**) measured by flow cytometry. One representative experiment (of 3) is shown with the number of adhered macrophages as percentage of input (*N* = 4–5), mean, and SD. Ns *p* ≥ 0.05, * *p* < 0.05, ** *p* < 0.01, *** *p* < 0.001, unpaired *t*-test.

**Figure 5 cancers-14-03602-f005:**
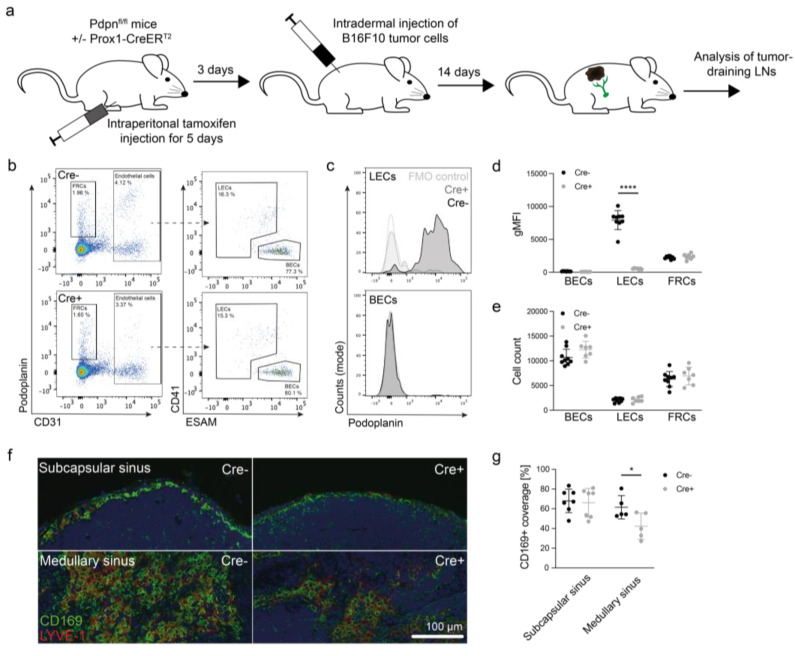
Lymphatic podoplanin regulates abundance of medullary sinus macrophages (MSMs) in tumor-draining lymph LNs. (**a**) Schematic representation of the experimental set-up. Pdpn^fl/fl^ x Prox1-CreER^T2^ mice were treated for 5 consecutive days with tamoxifen. Cre− littermates served as controls. Subsequently, B16F10 cells were implanted intradermally, tumor growth was monitored, and tumor-draining LNs (inguinal and axillary) were analyzed by flow cytometry on day 14. (**b**) Representative LEC gating in Cre− and Cre+ mice. Stromal cells (pregated as CD45− living singlets) were stained for podoplanin and CD31 to detect FRCs and endothelial cells, respectively. Subsequently, LECs were distinguished from BECs with a combination of CD41 (Itga2b) and ESAM. (**c**,**d**) Representative histogram (**c**) and quantification (**d**) of podoplanin expression in LN stromal cells (*N* = 9 Cre−/7 Cre+). (**e**) Number of LN stromal cells in tumor-draining LNs determined by flow cytometry (*N* = 9 Cre−/7 Cre+). (**f**,**g**) Representative images (**f**) and quantification (**g**) of CD169+ macrophages in subcapsular and medullary sinuses of tumor-draining LNs in Cre− and Cre+ mice (*N* = 7/group for subcapsular and 5/group for medullary sinus). (**h**–**j**) Representative flow cytometry gating ((**h**), pregated for CD11b+ living singlets), absolute quantification (**i**), and relative quantification (**j**) of LN macrophage subsets medullary cord macrophages (MCMs), MSMs and subcapsular sinus macrophages (SSMs) (*N* = 9 Cre−/6 Cre+). * *p* < 0.05, **** *p* < 0.0001, unpaired *t*-test.

## Data Availability

Raw scRNA-seq data of naive and tumor-draining LN LECs are accessible at ArrayExpress under the accession numbers E-MTAB-10434 and E-MTAB-11524.

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
