# Peer review of "Immunomodulatory Responses of Subcapsular Sinus Floor Lymphatic Endothelial Cells in Tumor-Draining Lymph Nodes"

_cancers, 2022, doi:10.3390/cancers14153602_

Round 1
Reviewer 1 Report
The manuscript has been significantly improved and is by far better than its previous version. I have 2 comments:
1) the Authors should carefully check the manuscript for spelling errors, e.g. The title: tumordraining or rather tumor draining/tumor-draining; Supplementary Table 1: ICH or IHC? Flow Cytomety – spelling error; lines 251, 255 - spaces missing
2) I would also consider placing Fig 1a vertically and 1b on the right, and 1c below 1b, then the Authors wouldn’t have empty space in Fig 1
Reviewer 2 Report
I acknowledged the authors for improving the overall quality of their manuscript. The authors incorporate key elements that highlight the role of podoplanin expression in tumor-draining lymph node. The authors also discussed more clearly about their previous studies and the difference in tumor-draining lymph nodes responses. It facilitates the reading and understanding . I hope that the authors are satisfied about the new version of their manuscript.
Reviewer 3 Report
This manuscript deals with the subcapsular sinus floor lymphatic endothelial cells (LECs) in tumour draining lymph nodes. The analysis is performed with single-cell RNA sequencing of LECs. Also, the effect of some pro-inflammatory cytokines such as TNF and IFN have been analysed.
This work is well organized and presented. The data are convincing and the conclusions are in line with the results shown. Also, the description of the limitations of the study are valuable.
The role of upregulation of CD31 induced by TNF in macrophage-endothelial cell adhesion (by homophilic adhesion) is not considered although highly conceivable and this may have a role in regulating the adhesion by positive and negative signals delivered between leukocytes and endothelial cells. at least some discussion or suggestions can be added.
Minor points (misprints)
resolution,.fLECs responded line 77
and toconfirm strongly reduced podoplanin line 405
This manuscript is a resubmission of an earlier submission. The following is a list of the peer review reports and author responses from that submission.
Round 1
Reviewer 1 Report
In this study, the authors described the response of LECs in tumor-draining lymph nodes using single-cell RNA sequencing. In particular, fLECs upregulate the expression of podoplanin compared to others LECs in tumor-draining lymph nodes. They showed that LECs changed their expression profile of adhesion molecules. The authors focused on 4 proteins: Podoplanin, Bst2, CD200 and tenascin. Whereas Podoplanin may play a role in adhesion of macrophages, CD200 and Bst2 have an opposite effect in vitro.
Fig 1b: The authors showed that most of the genes upregulated in tumor-draining lymph nodes are fLECs compared to cLECs and mLECs. It will be informative to present the volcano plots for the 3 populations of LECs in control and tumor-draining lymph nodes.
Podoplanin is higher expressed in fLECs in tumor-draining lymph nodes, but are also expressed at similar level in cLECs. What’ is the consequence of podoplanin deletion on the identity of fLECs, cLECs and mLECs?
Discussion: In vitro, Podoplanin deletion affect macrophage adhesions in untreated cells but not in TNFa stimulated cells suggesting that macrophages adhesion in inflammatory condition is independent of podoplanin (Fig 4e). This data is in contradiction with the conclusion of the authors that podoplanin regulates the adhesion of macrophages on LECs in tumor-draining lymph nodes. Podoplanin interacts with transmembrane co-receptors and also with CCL21. Do the authors observe a difference on the expression of others adhesives molecules or in chemokine expression in Pdpnfl/fl;Prox1-creERT2 mice?
It appears important to discuss why the authors think that podoplanin in LECs plays a role in adhesion of macrophages. The authors could provide more explanation and argument in the discussion.
Reviewer 2 Report
This manuscript uses scRNA-seq to transcriptionally characterize LN LECs in tumor-draining LNs. This is a well-written, original, and technically sound manuscript.
There are several minor concerns that I would like to see addressed, which are detailed below:
1) The manuscript length (20 pages) is more typical of a thesis, rather than a research article. All sections would greatly benefit from being shortened, as well as the title.
2) Figure 2d to i: please include a legend for red and blue on each graph.
3) Section 3.3 and Figure 3: no mention is made of data differences between 4,18 and 48h. Induced changes described in the text are difficult to match to Figure 3 at times and would benefit from a more clear explanation. Please include a legend for hours on each graph.
4) Figure 5b and h: FACS plot images are of low quality and barely legible.
5) Conclusions: What are the translational applications of these results? How could this be used to improve ICB efficacy?
6) Supplementary Figure S3d: please include a legend for black and grey on the graph.
Reviewer 3 Report
In the present study, E. Sibler et al. presented the results of single-cell RNA sequencing of lymphatic endothelial cells (LECs) residing in B16F10 melanoma-draining lymph nodes (LNs) as compared to normal LNs. They found that LECs lining the subcapsular sinus floor (fLECs) showed upregulated expression of podoplanin, CD200 and BST2 and that podoplanin might allow medullary macrophage adhesion to LN LECs.
There are unfortunately a couple of major problems in this study. First, the study is descriptive in nature, and the conclusion made from functional experiments are only speculative. Second, authors used only a single time point to examine transcriptional changes of LN LECs, i.e., 14 days, after tumor inoculation, which makes it difficult to judge whether the changes they observed were truly related to tumor growth. It is of note that intradermally injected B16 cells often grow rapidly, leading to substantial tumor tissue necrosis, which secondarily results in acute inflammation of the tumor tissue. Therefore, it is essential to verify that the changes they observed were actually induced by tumor growth but not by acute inflammation secondarily induced in the tumor tissue.
Reviewer 4 Report
In my opinion the manuscript should not be accepted for publication in Cancers, at least in the current form, it should be re-arranged. I have also concerns about the study design. A big part of the Results is just confirmation of the previous foundings(1st paragraph) or showing lack of changes. I also do not understand the study design. Why the Focus on Cd200 or Bst2 or Pdpn? Were they among top deregulated changes? The authors do not show the significance of their expression changes with referring to the list of all changed genes. My detailed comments are below:
- In my opinion the 1st paragraph of the Results section should be joined with Fig.2 into one Figure, as well as the two corresponding the Results sections. This is because it does not involve new results.
- I have concerns about presenting the Results in Fig2A as a graph. A simple table would be enough.
- In the Fig2. The authors focus on concrete genes but do not refer to their bioinformatic analyses of biological processes. They show some GO analyses in the supplemental materials (Fig. S2A). The general order in the figure should be first – to show top biological changed processes and second focus on the concrete genes. Additionally in the FigS2A, you say: Top 15 gene ontology terms for biological processes enriched among genes upregulated in fLECs from tumor-draining compared to control LNs. This is unclear for me, because many of the GO processes presented are involved in the negative regulation like: negative regulation of cell proliferation or angiogenesis that should be rather overrepresented in tumor variant. It should be clearly stated which are overrepresented and which underrepresented in tumor fLEC comparing to control.
- Why the authors concentrated on Pdpn, Cd200, Bst2 and Tnc in their further study? They are not top deregulated and they do not occur in their dotplot analysis.
- As for Materials and Methods section, it will be more transparent to provide Tables for antibodies and qPCR primers in the Supplemental Materials.
- In the Introduction section the authors describe different kinds of LN LECs with the emphasis on their location and function. It would be very helpful for the readers to provide a simple scheme of the arrangement of LECs in the lymph node and place it as Fig.1A.